# Free Radical Properties, Source and Targets, Antioxidant Consumption and Health

**Giovanni Martemucci** [1], **Ciro Costagliola** [2], **Michele Mariano** [3], **Luca D'andrea** [2], **Pasquale Napolitano** [4,*] and **Angela Gabriella D'Alessandro** [1]

[1] Department of Agricultural and Environmental Sciences, University of Bari Aldo Moro, Via G. Amendola, 70126 Bari, Italy; gmartem@libero.it (G.M.); angelagabriella.dalessandro@uniba.it (A.G.D.)

[2] Department of Neurosciences, Reproductive Sciences and Dentistry, University of Naples Federico II, 80131 Naples, Italy; ciro.costagliola@unimol.it (C.C.); dandrea.luca91@gmail.com (L.D.)

[3] Unità Operativa Complessa di Radiodiagnostica Universitaria—Policlinico di Bari, 70122 Bari, Italy; marmic37@libero.it

[4] Department of Medicine and Health Sciences "Vincenzo Tiberio", Università del Molise, 86100 Campobasso, Italy

[*] Correspondence: napolitano.pasquale1989@gmail.com

**Abstract:** Free radicals have acquired growing importance in the fields of biology and medicine. They are produced during many different endogenous and exogenous processes. Mitochondria are the main source of endogenous reactive oxygen species (ROS) produced at cell level. The overproduction of free radicals can damage macromolecules such as nucleic acids, proteins and lipids. This leads to tissue damage in various chronic and degenerative diseases. Antioxidants play a crucial role in the body's defense against free radicals. This review concerns the main properties of free radicals, their sources and deleterious effects. It highlights the potential role of the dietary supplementation of antioxidants and discusses unsolved problems regarding antioxidant supplements in the prevention and therapy of diseases.

**Keywords:** free radicals; reactive oxygen species; reactive nitrogen species; molecular targets; physiopathology; antioxidant supplementation

## 1. Introduction

In recent years, there has been growing interest in the prevention of disease, with particular reference to the role of free radicals. The biology and medicine of free radicals are expanding rapidly because they are profoundly relevant to health and disease and, therefore, quality of life. Free radicals were discovered more than a century ago, when it seemed that all oxidation reactions involving organic molecules were mediated by free radicals [1]. In the 1950s, free radicals were detected in biological systems and were proposed to be involved in human diseases [2]. In 1969, McCord and Fridovich [3] discovered the enzyme superoxide dismutase (SOD), demonstrating that living organisms have protective systems. This finding was sustained by subsequent discoveries of other antioxidant enzymes and antioxidant protein metabolites. It became clear that the deleterious effects of free radicals can be controlled by specific antioxidant systems. Most research on free radicals concerns oxygen radicals, which, together with some forms of active non-radical oxygen, are collectively called reactive oxygen species (ROS). In 1971, it was discovered [4] that reactive oxygen species are generated in cell metabolic respiration; later, Halliwell and Gutteridge [5] reported that ROS included free radical and non-radical derivatives of oxygen. In the 1980s, free radical nitric oxide ($^\bullet$NO) was identified in blood vessels and gave rise to studies on the biochemistry of reactive nitrogen species (RNS) [6]. More recently, it has been found that free radicals also play a positive biological role; ROS and reactive nitrogen species (RNS) operate in concert with reactive halogen species as part of

the cell immune response to infection by microorganisms [7]. The signaling functions of ROS and RNS are the latest major biological discovery on free radicals [8].

Here, we provide an overview of the properties, generation and molecular targets of free radicals and their implications in diseases. The potential role of antioxidant supplementation for health maintenance is also discussed.

## 2. Free Radicals

In the history of the Earth, the evolutionary selection of oxygen has depended on its ready availability, high energy of oxidation, easy distribution and solubility in biocomponents. However, there are negative aspects, since breathing pure oxygen (100% $O_2$, 1 atm) instead of air (20% $O_2$, 1 atm) is harmful to living organisms. Oxygen is essential for the life of aerobic organisms. The ability to use oxygen has enabled humans and animals to metabolize fats, proteins and carbohydrates to produce energy, albeit not without a cost since, paradoxically, the use of oxygen contributes to human aging and illness. Free radicals are products of normal cell metabolism. When cells use oxygen, the redox process generates free radicals, normally ROS and RNS [9].

Free radicals can be defined as molecular entities or molecular fragments, capable of independent existence (hence "free"). They contain one or more unpaired electrons in an outer atomic orbital or molecular orbital (hence "radical") [10]. The negative electrical charge of electron(s) may be counterbalanced by the positive nuclear charge of positrons, resulting in a neutral particle; otherwise, we have anion or cation radicals. The unpaired electron of a free radical is denoted by a point on the atom or group in which it predominantly resides, for example $^\bullet H$ (hydrogen atom), $^\bullet OH$ (hydroxyl), $^\bullet CH_3$ (methyl) and $^\bullet CH_2CH_3$ (ethyl). In oxygen radicals, the unpaired electron is located predominantly on an oxygen atom, e.g., superoxide ($O_2^{\bullet-}$), hydroxyl ($^\bullet OH$), peroxyl ($ROO^\bullet$). The odd number of electron(s) of a free radical makes it unstable, short-lived and highly reactive. This characteristic is responsible for chain reactions. Free radicals attempt to bond with other molecules, atoms or even individual electrons to create a stable compound. They either donate or accept an electron from other molecules, acting as oxidizing or reducing agents [10].

Reactive oxygen and nitrogen species may be radicals or non-radical compounds [11,12] (Table 1). Reactive oxygen species include the three chemical species of the Fenton/Haber-Weiss pathway ($O_2^{\bullet-}$ superoxide radical, $H_2O_2$ hydrogen peroxide and $HO^\bullet$ hydroxyl radical), products of the partial reduction of oxygen. Four-electron reduction of molecular oxygen leads to the formation of water without the generation of ROS, whereas one-electron reduction leads to the formation of the superoxide radical, hydrogen peroxide and the hydroxyl radical [13]. $O_2^{\bullet-}$ and $HO^\bullet$ have unpaired electrons in external orbitals and are therefore defined as free radicals, whereas $H_2O_2$ does not have unpaired electrons and is therefore not a radical. Non-radical derivatives of oxygen include hydrogen peroxide ($H_2O_2$), ozone ($O_3$) and singlet oxygen ($^1O_2$). Reactive nitrogen species are nitrogen-based radicals, including the three chemical species of the Beckman-Radi-Freeman pathway ($NO^\bullet$ nitric oxide, $ONOO^\bullet$ peroxynitrite and $NO2^\bullet$ nitrogen dioxide) [10,14]. Collectively, the radicals include superoxide ($O_2^{\bullet-}$), oxygen radical ($O_2^{\bullet\bullet}$), hydroxyl ($^\bullet OH$), alkoxy radical ($RO^\bullet$), peroxyl radical ($ROO^\bullet$), nitric oxide (or nitrogen monoxide) ($NO^\bullet$) and nitrogen dioxide ($NO_2^\bullet$) [11,12]. The non- radical species include hydrogen peroxide ($H_2O_2$), hypochlorous acid (HOCl), hypobromous acid (HOBr), ozone ($O_3$), singlet oxygen ($^1O_2$), nitrous acid ($HNO_2$), nitrosyl cation ($NO^+$), nitroxyl anion ($NO^-$), dinitrogen trioxide ($N_2O_3$), dinitrogen tetroxide ($N_2O_4$), nitronium (nitryl) cation ($NO^{2+}$), organic peroxides (ROOH), aldehydes (HCOR) and peroxynitrite ($ONOO^-$) [11,12]. These non- radical species are not free radicals, but they are readily trigger free radical reactions in living organisms [15]. The different types of free radicals vary widely in their reactivity; for example, the reactivity of ROS in decreasing order is: $HO^\bullet > O_2^{\bullet-} > H_2O_2$ [10]. The chemical reactivity of free radicals is directly associated with their potential to damage biological molecules. $^\bullet OH$ is much more reactive than the other species and reacts rapidly

with almost all chemical species, while $H_2O_2$, $NO^\bullet$ and $O_2^{\bullet-}$ react quickly with a few molecules. Other species, such as $RO_2^\bullet$, $NO_3^\bullet$, $RO^\bullet$, HOCl, $NO_2^\bullet$, $ONOO^-$, $NO_2^\bullet$ and $O_3$, show intermediate reactivities. $RO^\bullet$ is more reactive than $ROO^\bullet$, the main products of lipid peroxidation. Regarding RNS, the reactivity of $NO_2^\bullet$ is between those of $NO^\bullet$ and $ONOO^-$. $NO^\bullet$ has a rather low chemical reactivity, and its toxicity is therefore not high. However, when it reacts with $O_2^{\bullet-}$, it produces a highly toxic species $ONOO^-$ capable of damaging lipids, proteins and DNA. The term ROS is used to include not only radicals but also non-radical species. However, biological systems are mainly damaged by radicals generated from oxygen.

**Table 1.** Main reactive oxygen and nitrogen species generated during metabolism [11,12].

| Reactive Oxygen Species (ROS) | | | Reactive Nitrogen Species (RNS) | | |
|---|---|---|---|---|---|
| Name | Symbol | Half-Life (s) | Name | Symbol | Half-Life [1] |
| Radicals | | | | | |
| Superoxide | $O^{\bullet-}$ | $10^{-6}$ s | Nitric oxide | $NO^\bullet$ | s |
| Hydroxyl | $^\bullet OH$ | $10^{-10}$ s | Nitrogen dioxide | $NO_2^\bullet$ | s |
| Hydroperoxyl | $HO_2^\bullet$ | s | Nitrate radical | $NO_3^\bullet$ | s |
| Peroxyl | $ROO^\bullet$ | 17 s | | | |
| Alkoxyl | $RO^\bullet$ | $10^{-6}$ s | | | |
| Organic hydroperoxide | ROOH | Stable | | | |
| Non-radicals | | | | | |
| Hydrogen peroxide | $H_2O_2$ | Stable | Nitrous acid | $HNO_2$ | s |
| Ozone | $O_3$ | s | Nitrosonium cation | $NO^+$ | s |
| Singlet oxygen | $(^1O_2Dg)$ | $10^{-6}$ s | Nitroxyl anion | $NO^-$ | s |
| Hypochlorous acid | HOCl | Stable (min) | Peroxynitrite | $ONOO^-$ | $10^{-3}$ s |
| Peroxynitrite | $ONOO^-$ | $10^{-3}$ s | Dinitrogen trioxide | $N_2O_3$ | s |
| | | | Dinitrogen tetroxide | $N_2O_4$ | s |
| | | | Peroxynitrous acid | ONOOH | Fair stable |
| | | | Nitryl chloride | $NO_2Cl$ | s |

[1] s: seconds; min: minutes; the half-life of some radical changes in relation to environmental medium; for example, $NO^\bullet$ in a saturated solution of air may have a half-life of a few minutes.

The abundance of free radicals in decreasing order is reactive oxygen species > reactive nitrogen species > reactive sulfur species [12,16] (Table 2).

**Table 2.** List of the common reactive oxygen species, nitrogen species and sulfur species of biological interest [12,16].

| Reactive Oxygen Species (ROS) | | Reactive Nitrogen Species (RNS) | | Reactive Sulfur Species (RSS) | |
|---|---|---|---|---|---|
| Name | Symbol | Name | Symbol | Name | Symbol |
| Radicals | | | | | |
| Oxygen (bi-radical) | $O_2\cdot\cdot$ | Nitric oxide | $NO^\bullet$ | Thiyl radical $S^\bullet$ | $RS^\bullet$ |
| Hydroxyl | $^\bullet OH$ | Nitrogen dioxide | $NO_2^\bullet$ | | |
| Hydroperoxyl | $HO_2^\bullet$ | Nitrate radical | $NO_3^\bullet$ | | |
| Carbonate | $CO_3^{\bullet-}$ | | | | |
| Peroxyl | $ROO^\bullet$ | | | | |
| Alkoxyl | $RO^\bullet$ | | | | |
| Carbon dioxide radical | $CO_2^{\bullet-}$ | | | | |

**Table 2.** *Cont.*

| Reactive Oxygen Species (ROS) | | Reactive Nitrogen Species (RNS) | | Reactive Sulfur Species (RSS) | |
|---|---|---|---|---|---|
| Name | Symbol | Name | Symbol | Name | Symbol |
| Non-radicals | | | | | |
| Hydrogen peroxide | $H_2O_2$ | Nitrosyl cation | $NO+$ | Hydrogen sulfide | $H_2S$ |
| Ozone | $O_3$ | Nitrous acid | $HNO_2$ | Disulfide | RSSR |
| Singlet oxygen | $^1O_2$ | Nitroxyl anion | $NO^-$ | Disulfide-S-monoxide | RS(O)SR |
| Hypobromous acid | HOBr | Dinitrogen trioxide | $N_2O_3$ | Disulfide-S-dioxide | RS(O)2SR |
| Hypochlorous acid | HOCl | Dinitrogen tetroxide | $N_2O_4$ | Sulfenic acid | RSOH |
| Hypoiodous acid | HOI | Dinitrogen pentoxide | $N_2O_5$ | Thiol/sulfide | RSR |
| Organic peroxides | ROOH | Alkyl peroxynitrites | ROONO | | |
| Peroxynitrite | $ONOO^-$ | Alkyl peroxynitrates | $RO_2ONO$ | | |
| Peroxynitrate | $O_2NOO^-$ | Nitryl chloride | $NO_2Cl$ | | |
| Peroxynitrous acid | ONOOH | Peroxyacetyl nitrate | $CH_3C(O)OONO_2$ | | |
| Peroxomono-carbonate | $HOOCO_2^-$ | | | | |
| Carbon monoxide | CO | | | | |

## 2.1. Properties of Free Radicals

### 2.1.1. Superoxide Anion Radical ($O_2^{\bullet-}$)

The superoxide anion ($O_2^{\bullet-}$) is a reduced form of molecular oxygen created by receiving an electron in a $\pi^*$ antibonding orbital [10]. With only one unpaired electron, superoxide is less radical than $O_2$, and despite its super name, its reactivity with biomolecules is not very sustained [17]. The addition of another electron to $O_2^{\bullet-}$ produces $O_2^-$, the peroxide ion, a non-radical (no unpaired electrons) with a weaker oxygen–oxygen bond. The addition of another two electrons to $O_2^{2-}$ completely eliminates the bond, producing two $O_2^-$ (oxide ions). In biology, the two-electron reduction product of $O_2$ is $H_2O_2$, and the four-electron product is water. It is mostly produced in the mitochondrial electron transport chain in the course of oxidative phosphorylation, which produces adenosine triphosphate (ATP) [18,19]. The superoxide anion can be produced by enzymic or non-enzymic activity, by the direct transfer of electrons to an oxygen molecule [20] or by photochemical means [21]; in biological systems, it is the main precursor of highly reactive species such as $HO^\bullet$, $^1O_2$, $CO_3^{\bullet-}$, $ONOO^\bullet$, HOCl and $GSSG^{\bullet-}$ (glutathione disulfide) [10,22]. The enzymes that produce superoxide include oxygenases dependent on cytochrome P450 and xanthine oxidase dependent on lipoxygenase, cyclooxygenase and nicotinamide adenine dinucleotide phosphate (NADPH) oxidase [20]. A superoxide radical, as a moderately reactive free radical, can react with another superoxide radical to produce hydrogen peroxide ($H_2O_2$), which can be reduced to water or partially reduced to the extremely reactive hydroxyl radical ($HO^\bullet$). Dismutation of the superoxide radical can be spontaneous or catalyzed by enzymes known as superoxide dismutases. The formation of $HO^\bullet$ is possible by the decomposition of $H_2O_2$, catalyzed by transition metal ions in the lower valence state, such as $Fe^{2+}$ or $Cu^+$ (Fenton reaction), or by the reaction of $H_2O_2$ with a superoxide radical (Haber–Weiss reaction); oxidized transitional metals from the Fenton reaction may be re-reduced by $O_2^{\bullet-}$ [23].

Since superoxide is a highly reactive free radical, it can damage molecules (DNA, proteins and lipids) [10]. It may be generated by the immune system to kill invading microorganisms; phagocytes, such as neutrophils, monocytes, macrophages, mast cells and dendritic cells, are mobilized by chemotaxis to the site of bacterial infection and mediate damage through their surface receptors. The phagocytosed bacteria are killed by a process involving $O_2^{\bullet-}$ [17].

### 2.1.2. Hydroxyl Radical (HO$^\bullet$)

The hydroxyl radical (HO$^\bullet$) is, chemically, the most reactive free radical formed in vivo. It is formed by the Fenton reaction, in which free iron (Fe$^{2+}$) reacts with hydrogen peroxide (H$_2$O$_2$), and by the Haber–Weiss reaction of superoxide with ferric iron (Fe$^{3+}$), producing Fe$^{2+}$. The reaction is not limited to iron, but it may involve several other ions (Cu$^{2+}$, Fe$^{3+}$, Ti$^{4+}$ and Co$^{3+}$), which can be recycled by interaction with superoxide anion to form O2 [24]. It is estimated that a cell produces around 50 hydroxyl radicals per second [25]; since hydroxyl radicals have the highest one-electron reduction potential (2310 mV), they can react with anything in living organisms with rate constants from 109 to 1010/M/s [26] and are considered the most harmful free species, since they attack any molecule less than a few nanometers from where they are generated.

The hydroxyl radical reacts strongly with most organic and inorganic molecules (DNA, proteins, lipids, amino acids, sugars, vitamins and metals) faster than its speed of generation [27]. These reactions involve the abstraction of hydrogen and the addition and transfer of electrons [10,11]. In saturated compounds, a hydroxyl radical abstracts a hydrogen atom from the weaker C–H bond to produce a free radical [26]. The resulting radicals may react with oxygen and generate other free radicals. Hydroxyl radicals are easily added to double bonds. All mitochondrial enzyme proteins are susceptible to inactivation by HO$^\bullet$, while all amino acid residues of proteins can be oxidized by HO$^\bullet$ [28]. It is estimated that ·OH is responsible for 60–70% of the tissue damage caused by ionizing radiation [29]. Hydroxyl radicals are also involved in disorders, such as cardiovascular disease [30] and cancer [31].

### 2.1.3. Peroxyl Radical (ROO$^\bullet$)

The alkoxyl (RO$^\bullet$) and peroxyl (ROO$^\bullet$) radicals are oxygen-centered organic radicals. They tend to accept electrons and then undergo reduction, having highly positive reduction potentials (1000 to 1600 mV) [32]. Peroxyl and alkoxyl radicals can be generated by the decomposition of alkyl peroxides (ROOH) induced by heat, radiation or a reaction with transition metal ions and other oxidants capable of subtracting hydrogen [32]. They can also be generated by the oxidation of proteins and nucleic acid [33]. These carbon-centered radicals react directly with biological molecules, such as DNA and albumin -SH-groups. They can abstract hydrogen from other molecules that have a lower standard reduction potential, as observed in the propagation phase of lipid peroxidation. The alkyl radical formed by this reaction may react with oxygen to form another peroxyl radical, resulting in a chain reaction. The RO$^\bullet$ radicals formed by the reduction of peroxides are significantly more reactive than ROO$^\bullet$ but less reactive than $^\bullet$OH [34]. ROO$^\bullet$ may diffuse to remote parts of cells. Their half-lives are of the order of seconds, and they are generally less reactive than HOO$^\bullet$ when R is an alkyl or an alkenyl group [35]. Some peroxyl radicals cleave, releasing superoxide anion, or react with each other to generate singlet oxygen [10].

### 2.1.4. Hydroperoxyl Radical (HO$_2$$^\bullet$)

HO$_2$$^\bullet$, usually termed hydroperoxyl radical or perhydroxyl radical, is the simplest form of a peroxyl radical, produced by the protonation of the superoxide anion radical or by the decomposition of hydroperoxide; approximately 0.3% of superoxide present in the cell cytosol exists in the protonated form [36]. The hydroperoxyl radical produces H$_2$O$_2$, which can react with active redox metals, including iron and copper, to trigger Fenton or Haber–Weiss reactions. The hydroperoxyl radical can also extract hydrogen atoms from NADH or glyceraldehyde-3-phosphate dehydrogenase–NADH, forming H$_2$O$_2$ [37]. Its reactions are slower than HO$^\bullet$ but competitive with organic peroxyl radicals. The hydroperoxyl radical plays an important role in the chemistry of lipid peroxidation. It is a much stronger oxidant than superoxide anion due to its ability to extract hydrogen atoms from linoleic, linolenic and arachidonic fatty acids, suggesting a role in the initiation of lipid oxidation [10,37].

### 2.1.5. Hydrogen Peroxide ($H_2O_2$)

Hydrogen peroxide can be generated by the dismutation of $O_2^{\bullet-}$ or by the direct reduction of $O_2$, and it is mainly produced by enzyme reactions [38]. The presence of oxidases (urate oxidase, glucose oxidase, D- amino acid oxidase) may lead to the direct synthesis of hydrogen peroxide by the transfer of two electrons to molecular oxygen; these enzymes are found in microsomes, peroxisomes and mitochondria [38]. Hydrogen peroxide is liposoluble and can therefore diffuse through the cell membrane. Being weakly reactive, this non-free-radical cannot readily oxidize most lipids, proteins and nucleic acids. The threat posed by $H_2O_2$ lies in its conversion to the hydroxyl radical ($HO^\bullet$) by homolytic fission, induced by UV or by the interaction with transition metal ions (Fenton reaction) [39]. Hydrogen peroxide may produce singlet oxygen through a reaction with a superoxide anion or with HOCl or chloramines in living systems [22]. The direct action of $H_2O_2$ involves an attack on the structure of heme proteins with the release of iron, enzyme inactivation and oxidation of DNA, lipids, -SH groups and keto-acids [11].

### 2.1.6. Molecular Oxygen ($O_2^{\bullet\bullet}$) and Singlet Oxygen ($^1O_2$)

In the evolutionary history of the Earth, oxygen appeared two billion years ago, what we call the ''Great Oxidation Event'', by virtue of the photosynthesis of cyanobacteria, which used solar energy to split water [40]. Oxygen, a metabolic by-product, was released into the atmosphere [25], where it formed the ozone ($O_3$) that shields the Earth from the radiation. Oxygen removed ferrous iron ($Fe^{2+}$) from aqueous environments by forming deposits of insoluble ferric complexes, leaving only traces of soluble iron in sea and river water [25]. Since animals, including humans, need $O_2$, a toxic mutagenic gas for the mitochondria to efficiently produce energy, only advanced antioxidant defenses allow them to survive. In fact, all aerobic organisms, including plants, aerobic bacteria, animals and humans, suffer damage if exposed to higher-than-normal concentrations of $O_2$ [10]. This means that their antioxidant defenses are limited. According to the theory of superoxide toxicity, $O_2$ toxicity is due to excessive superoxide radical formation [17]. From the biological point of view, molecular oxygen, in its diatomic ($O_2$) ground state, is a bi-radical because it contains two unpaired electrons, each of which is located in a different $\pi$ antibonding orbital. It is indicated as "triplet oxygen" because the spin of these electrons has three possible alignments with an external field [41]. Triplet oxygen, the more abundant form of oxygen, is the common oxygen we breathe. It carries a "spin restriction" against reacting with most organic molecules. Molecular oxygen is not very reactive because its electrons are in the lowest energy configuration.

When the two unpaired electrons from triplet oxygen enter two different orbitals, the result is a powerful oxidant named singlet oxygen ($^1\Delta g$, $^1O_2$) [42]. The $^1\Delta g$ state, which is 92 kJmol$^{-1}$ above the ground state, carries an empty $\pi^*$ orbital where it can accommodate a pair of electrons. This ability gives singlet oxygen strong acidic properties. It is therefore a strong electrophile, which reacts with reagents that have high electron density regions, oxidizing them.

Photosensitizers, such as hematoporphyrins, riboflavin and myoglobin, may form singlet oxygen from triplet oxygen in the presence of light by two basic types of photo-oxidation [21]. In the type I reaction, the photosensitizer absorbs light, enabling the excited triplet to react directly with the substrate; while in the type II reaction, it first interacts with the molecular oxygen ground-state ($^3O_2$) to produce $^1O_2$, and the excited triplet returns to its ground state. The speed of the type I or II reaction depends on sensitizer type [21,43] and on the substrate and concentrations of substrate and oxygen in the reaction environment. Additionally, $^1O_2$ is produced in vivo by the activation of eosinophils, macrophages and neutrophils [43] and by the enzyme reactions and activities of different peroxidases [42].

Singlet oxygen is very reactive because the "spin restriction" is removed, allowing the species to react as an electrophilic oxidant [21,44] and making it a potential aggressor when it is produced inside the cell [45]. This is indicated especially by its ability to damage DNA, components of guanine and nucleic acids, leading to toxic and mutagenic effects and tissue damage [43]. It is also involved in the oxidation of cholesterol [46] and proteins with high electron density amino acid residues, such as cysteine, methionine, tryptophan, tyrosine and histidine [28]. Singlet oxygen can also play a role in generating cell signals to modify gene expression [43] and can be used to fight cancer cells and various pathogens such as microbes and viruses [22].

### 2.1.7. Ozone ($O_3$)

In the history of the Earth, ozone was formed from $O_2$ by the action of high energy electromagnetic radiation and electrical discharges [25]. It is slightly less reactive than $HO^\bullet$ and a much stronger oxidizing agent than oxygen [46]. It can form free radicals by oxidizing biological molecules and causes oxidative damage to lipids [47], proteins and nucleic acids [48]. Ozone also plays an important role in inflammatory processes [49].

### 2.1.8. Hypochlorous Acid (HOCl)

Hypochlorous acid (HOCl) is a highly reactive species involved in oxidation reactions and chlorination of the protein and lipid components. It is generated by hydrogen peroxide and the chloride anion in a reaction catalyzed by myeloperoxidase in macrophages and neutrophils at sites of inflammation [50]. It can oxidize thiols and other biological molecules, including ascorbate, urate, pyridine nucleotides and tryptophan [50]. HOCl chlorinate compounds such as amines to chloramines, residues of tyrosyl to ring chlorinated products, cholesterol and unsaturated lipids to chlorohydrins and may also chlorinate DNA [51].

### 2.1.9. Carbonate Radical Anion ($CO_3^{\bullet-}$)

The carbonate radical anion ($CO_3^{\bullet-}$) may be produced by the radiolysis of aqueous solutions of bicarbonate/carbonate [52]; it can also be formed when $^\bullet OH$ reacts with carbonate or bicarbonate ions. Levels of bicarbonate are high (25 mM) in blood plasma, enabling the reaction [53]. Although not as strong an oxidizing agent as the hydroxyl radical, the carbonate radical anion is a strong one-electron oxidant that acts by electron transfer and hydrogen abstraction [33]. It has a much longer half-life than $^\bullet OH$ and can therefore spread further and oxidatively modify distant cell targets. A wide variety of biomolecules can be oxidized by $CO_3^{\bullet-}$. Regarded as a major oxidant of proteins and nucleic acids, it oxidizes DNA guanine bases by a one-electron transfer process that leads to the formation of stable guanine oxidation products [54]. The carbonate radical anion has been proposed as a key mediator of oxidative damage derived from peroxynitrite production [33,55], xanthine oxidase turnover and superoxide dismutase activity [56]. It is known to play an important role in the modification of selective amino acids in proteins under conditions of oxidative stress, aging and inflammation [57]. The kinetics of tyrosine nitration in the presence of $CO_2$ suggest a specific role of $CO_3^{\bullet-}$ in MnSOD nitration by peroxynitrite [58]. The nitration of tyrosine has been observed in neurodegenerative conditions, cardiovascular disorders and diabetes [59].

### 2.1.10. Nitric Oxide ($NO^\bullet$)

Nitric oxide ($NO^\bullet$), nitrogen dioxide ($NO_2^\bullet$) and peroxynitrite ($ONOO^-$), as well as non-radicals such as nitrous acid $HNO_2$ and $N_2O_4$ (dinitrogen tetroxide), are included in the collective term reactive nitrogen species (RNS). Nitric oxide or nitrogen monoxide ($NO^\bullet$) is a free radical with a single unpaired electron. The chemical reactivity of $NO^\bullet$ is rather limited, and consequently its direct toxicity is less than that of ROS. However, it reacts with $O_2^{\bullet-}$, producing peroxynitrite anion ($ONOO^-$) [55], a very damaging species for proteins, lipids and DNA [60]. Nitric oxide also reacts with molecular oxygen and nitrogen to form nitrogen dioxide or dinitrogen trioxide, both toxic oxidizing and nitrosating agents [55].

Nitric oxide is generated in biological tissues by specific nitric oxide synthases [61], through the reaction of $H_2O_2$ with arginine [62] or through the decomposition of S-nitroso thiols in the presence of metal ions [63].

Nitric oxide is soluble in water and fat, and it therefore diffuses readily through the cytoplasm and plasma membrane. If human blood plasma is exposed to $NO^\bullet$, ascorbic acid and uric acid concentrations become depleted and lipid peroxidation is triggered [10]. Nitric oxide-derived species in cell membranes and lipoproteins react quickly with fatty acids and lipid peroxyl radicals during lipid oxidation, generating oxidized and nitrated products of free lipids and esterified cholesterol [64]. Nitric oxide is also involved in many physiological processes, such as neuro-transmission, relaxation of smooth muscle, vasodilation and regulation of blood pressure, gene expression, defense mechanisms, cell function and regulation of inflammatory and immune mechanisms, as well as in pathological processes such as neurodegenerative disorders and heart diseases [14].

### 2.1.11. Nitrogen Dioxide ($NO_2^\bullet$)

Unlike nitrous oxide ($N_2O$), nitrogen dioxide ($NO_2^\bullet$) can be considered a free radical because the electrons are not paired. It is formed by the reaction of the peroxyl radical and NO in polluted air and smoke [65]. Nitrogen dioxide is a moderately strong oxidant, with reactivity between those of $NO^\bullet$ and $ONOO^-$ [10]. $NO_2^\bullet$ reacts with organic molecules at rates ranging from ~104 to 106 M/s, depending on pH [33]. Two $NO_2^\bullet$ radicals can be dimerized to the highly reactive dinitrogen tetroxide ($N_2O_4$). Nitrogen dioxide can affect antioxidant mechanisms, causing the oxidation of ascorbic acid, which leads to lipid peroxidation and free radical production [66].

### 2.1.12. Peroxynitrite ($ONOO^-$)

Peroxynitrite ($ONOO^-$) is formed by the reaction of nitric oxide and superoxide anion. It is highly toxic and can react directly with $CO_2$ to form other highly reactive nitroso-peroxo-carboxylates ($ONOOCO_2^-$) or peroxynitrous acid ($ONOOH$), which may undergo further homolysis to form $^\bullet OH$ and $NO_2^\bullet$ or rearrange to form $NO_3$ [67]. Peroxynitrite diffuses readily across cell membranes [24]; it can oxidize lipids, methionine residues and tyrosine in proteins and DNA to nitroguanine [11,60]. It acts as an oxidant in a similar way to the hydroxyl radical [10]. Nitrotyrosine residues are considered markers of cell damage induced by peroxynitrite and have been associated with tissues aging [11,24]. Peroxynitrite causes tissue injury and oxidizes low-density lipoprotein (LDL); it seems to be generated at sites of inflammation [24,66].

### 2.1.13. Reactive Sulfur Species

Sulfur is very abundant in nature and in the human body, and it has been implicated in the origin of life [68]. Organic derivatives of sulfur can form thiols (−2), disulfides (−1), sulfenic acids or sulfoxides (0), sulfinic acid (+2) and sulfonic acids (+4). By analogy with ROS and RNS, these compounds are identified as reactive sulfur species (RSS) [16]. Thiols can generate free radicals. In the presence of traces of transition metal ions, thiol compounds are oxidized to thiyl radicals ($RS^\bullet$) and reactive oxygen species. Pro-oxidative action takes place by means of reduction of transition metals such as $Fe^{3+}$ to $Fe^{2+}$, leading to the formation of thiyl radicals and the generation of a superoxide radical anion. The biochemistry of thiols, hydrogen sulfide (H2S) and its sulfane sulfur derivatives enables roles in protein structure/folding, cell redox homeostasis, signaling, metal ligation, cell protection, enzymology, metabolism and mitochondrial function [69]. Humans and animals are continuously exposed to many exogenous thiols and related disulfides. Thiol compounds can be found in food, contaminants and products in which sulfur-containing substances are breaking down.

Hydrogen sulfide ($H_2S$) is the hydrogenated sulfur compound with the lowest oxidation state ($-2$). It is slightly hydrophobic and soluble in lipid membranes, which it crosses rapidly, diffusing between compartments. $H_2S$ exerts many physiological activities with potential health benefits [68]. It is mostly synthesized enzymatically, but also non-enzymatically in mammalian tissues [70], and it is produced via different pathways, in mitochondria [71], the kidneys and the brain [72]. Hydrogen sulfide has traditionally been considered toxic to mammals because of its inhibitory effect on cytochrome c oxidase, interrupting oxidative phosphorylation [73]. Since the identification of nitric oxide (NO) and carbon monoxide (CO) as gasotransmitters, $H_2S$ has been recognized as the third gasotransmitter [69]. Similar to NO and CO, $H_2S$ was considered to regulate various physiological and pathological processes.

Recently, the biological action and signaling of hydrogen sulfide [68] have stimulated interest in species related to $H_2S$ and/or as possible mediators/biological effectors derived from it. The biological effects mediated by $H_2S$ have mainly been attributed to the persulfidation of proteins, as shown by its vasorelaxant effect mediated by the activation of ATP-sensitive $K^+$ [74]. $H_2S$ also blocks the generation of mitochondrial ROS through the induction of p66Shc persulfidation [75] and reduces the advanced toxicity of glycation end products through persulfidation of their receptor [76]. It is involved in inflammatory processes [77], inhibiting leukocyte adherence [78] and in carcinogenesis [79]. Polysulfides protect neurons from oxidative stress through the activation of the Keap1/Nrf2 (Kelch-like ECH associated protein 1/transcription nuclear factor erythroid 2-related factor 2) system and also induce neurite growth [80]. $H_2S$ appears to inhibit atherogenesis and platelet aggregation [81], and it has been shown to protect against ischemia-reperfusion damage by the preservation of mitochondrial function [82]. It is stressed that, in biological systems, sulfur species, such as hydrogen sulfide, disulfides, hydropersulfides, dialkyltrisulfides and thiols, are all in dynamic equilibrium [83]. Hydropersulfide, not $H_2S$, has been proposed as a new product of interest in signaling research; in many cases, $H_2S$ could be a marker of its presence [84].

## 2.2. Generation of Free Radicals

Radicals can be formed by mechanisms other than the addition of a single electron to a non-radical. They can form by homolytic fission, when a covalent bond (C–C, C–H or C–O) is broken and one electron from the bonding pair remains on each atom. These covalent bonds are difficult to break; others are more easily broken, such as disrupted disulfide bonds that generate sulfur radicals [85], whereas the O–O bond in $H_2O_2$ is divided by exposing it to ultraviolet light, generating $^\bullet OH$. Sources of free radicals may be endogenous or exogenous (Figure 1). Endogenous sources, generated during normal metabolism, include different cell organelles, such as mitochondria, peroxisomes and endoplasmic reticulum, many enzyme activities, fatty acid metabolism and phagocytic cells [86]. Exogenous sources include radiation X-rays, $\gamma$-rays, ultraviolet A, visible light in the presence of a sensitizer, chemical reagents such as heavy or transition metals (e.g., Cd, Hg, Pb, As, metal ions such as $Fe^{2+}$ and $Cu^+$), HONOO, ozone, $N_2O_2$, deoxyosones, ketamine, $H_2O_2$, HOCl and HOBr, cooking (smoked meat, used cooking oil), high temperatures, environmental pollutants (aromatic hydrocarbons, pesticides, polychlorinated biphenyls, dioxins and many others), microbial infections, drugs and their metabolites [9,28,87].

**SOURCE OF FREE RADICALS**

| Exogenous | Endogenous |
|---|---|
| • Air and water pollution | • Cells (*neutrophilis, eosinophils,…*) |
| • Tobacco smoke | • Enzymes (*NO synthase, xanthin oxidase, NADPH oxidase, lipo-oxigenase, …*) |
| • Heavy metals | • Mitochondrial chain |
| • Transition metals | • Endoplasmatic reticulum oxidation |
| • Pesticides | • Cytochrome P450 |
| • High temperature | • Diseases |
| • UV irradiation | |
| • γ irradiation | |
| • Drugs | |
| • Cooking (*smoked meat, used oil*) | |

**FREE RADICALS**

**ROS**: $O_2^{\bullet-}$, $^\bullet OH$, $HO_2^\bullet$, $H_2O_2$, $RO_2$, $HOCl$ …

**RNS**: $NO^\bullet$, $NO_2^\bullet$, $ONOO^-$, $HNO_2$, $RONOO$, $N_2O_3$ …

**Figure 1.** Source of free radicals generation. Either endogenous or exogenous sources generate ROS ($O_2^{\bullet-}$, superoxide anion; $^\bullet OH$, hydroxyl; $HO_2^\bullet$, hydroperoxyl; $H_2O_2$, hydrogen peroxide; $RO_2$, peroxyl; $HOCl$, hypochlorous acid); RNS ($NO^\bullet$, nitric oxide; $NO_2^\bullet$, nitrogen dioxide; $ONOO^-$, peroxynitrite; $HNO_2$, nitrous acid; RONOO, alkylperoxynitrates; $N_2O_3$, dinitrogen trioxide). Endogenous free radicals are produced by the activation of immune cells (eosinophils, neutrophils, etc.) to fight bacteria and other invaders, by the mitochondrial respiratory chain, by enzymatic activity (xanthin oxidase, NADPH oxidase, lipo-oxygenase, NO synthase, etc.) and by various pathological disorders and diseases. Exogenous free radicals arise from air and water pollution, cigarette smoke, heavy metals or transition, drugs, industrial solvents, radiation and high temperatures.

### 2.2.1. Mitochondria

All the cells of the human body rely on adenosine triphosphate (ATP) to store and transport chemical energy. The body uses molecular oxygen to produce energy via oxidative phosphorylation in mitochondria. Mitochondria generate more than 90% of ATP by oxidative phosphorylation [88], consuming about 85% of the oxygen requirements of the cell to do so. Most of the oxygen is reduced to water, and a small proportion is converted to free radicals. The phosphorylation unit combines oxygen and hydrogen to produce $H_2O$ and ATP molecules. The oxidative unit consists mainly of a series of protein complexes in the inner mitochondrial membrane (IMM), known as the respiratory or electron transfer chain (ETC). Hydrogen atoms are known as reducing equivalents. The passage of hydrogen atoms along the respiratory chain is equivalent to the passage of electrons through sequential redox reactions along protein complexes I-IV of the ETC [89], where $O_2$ is reduced to $H_2O$. The production of ATP by oxidative phosphorylation associated with the ETC has an energy loss in the form of electrons [90], which determines the production of free radicals.

In eukaryotic organisms, over 90% of ROS are produced by the mitochondrial ETC as a by-product of respiration [91]. A quantity of ROS are also produced by the ETC in the plasma [92], nuclear [93] and endoplasmic reticulum [94] membranes.

Reactive oxygen species generated as by-products of mitochondrial electron transfer mainly include the superoxide radical anion and hydrogen peroxide. A multielectron reduction of $O_2$ is carried out by protein complexes in the ETC. By virtue of its electron configuration (two unpaired electrons in the outer shell), the oxygen molecule is not very reactive [41] and consequently tends to accept electrons one at a time. If $O_2$ accepts a single electron, the electron must enter an antibonding orbital, producing the superoxide radical $O_2^{\bullet-}$. A two-electron reduction of $O_2$, with the addition of $2H^+$, generates hydrogen peroxide ($H_2O_2$). A one-electron reduction of $H_2O_2$ forms a hydroxyl radical ($HO^{\bullet}$) and a hydroxyl anion $HO^-$. Water is formed after the electron and proton addition to $HO^{\bullet}$. Although the ETC is a highly efficient system, the redox reactions predispose electron vectors to reactions with molecular oxygen. Thus, up to 2% of electrons leak along the ETC and react directly with oxygen in a one-electron reduction to produce a superoxide (radical anion) instead of a water molecule [95]. About 5% of the oxygen consumed by living organisms can be converted to $O_2^{\bullet-}$ by mitochondria under physiological conditions [87]. The production of $O_2^{\bullet-}$ in mitochondria is estimated to be approximately 2 to 3 nmol/min per mg of protein [20], confirming its importance as the main source of this radical in living organisms.

Mitochondria are the most significant intracellular source of $O_2^{\bullet-}$. An $O_2^{\bullet-}$ concentration 5 to 10 times greater has been estimated in mitochondria than in the nuclear space or the cytosol [91]. Ubiquinone links complex I with III and II with III and is regarded as a major player in the formation of $O_2^{\bullet-}$. The oxidation of ubiquinone proceeds in a set of reactions known as the Q-cycle, and the unstable semiquinone is responsible for $O_2^{\bullet-}$ formation [44]. The transfer of electrons from complex I or II dehydrogenase to coenzyme Q or ubiquinone (Q) leads to the formation of a reduced form of coenzyme Q ($QH_2$) that regenerates coenzyme Q via an unstable intermediate semiquinone anion $Q^{\bullet-}$. The latter transfers electrons to molecular oxygen, leading to the formation of superoxide radical [44]. Since the generation of superoxide is not enzymic, most ROS production will be linked to the higher metabolic rate.

Additionally, mitochondrial superoxide is generated by electron-transfer during fatty acid oxidation, by glycerol-3-phosphate dehydrogenase and other IMM-associated oxidoreductases [96]. The superoxide anion ($O_2^{\bullet-}$) serves as a ROS precursor. Most $O_2^{\bullet-}$ is readily metabolized to non-radical $H_2O_2$ by superoxide dismutase (SOD) or non-enzyme mechanisms [97]. The subsequent Haber–Weiss reaction of $H_2O_2$ and $O_2^{\bullet-}$ [98], or $Fe^{2+}$- (or $Cu^{2+}$)-driven Fenton cleavage of $H_2O_2$ [99], may generate the highly reactive hydroxyl radical ($^{\bullet}OH$).

The $H_2O2$ produced is in its optimum state for respiration, characterized by a high degree of reduction of the electron carriers and a limiting supply of adenosine diphosphate (ADP) [100]. An additional source of $H_2O_2$, not related to breathing, is situated on the external mitochondrial membrane [101], where the oxidative deamination of biogenic amines by monoamine oxidases is associated with the direct two-electron reduction of $O_2$ to $H_2O_2$. The hydrogen peroxide produced during the oxidative deamination of catecholamines may be involved in neurodegenerative disorders such as Parkinson's and Alzheimer's diseases, presumably through oxidative damage to the mitochondrial membrane [102]. The factors that control the ETC generation of ROS in vivo are not fully understood. Antioxidant enzymes can eliminate ROS. SODs convert $O_2^{\bullet-}$ to $H_2O_2$, and many enzymes, such as catalase, glutathione peroxidase and peroxiredoxin 3, remove $H_2O_2$ [19]. Moreover, the signaling capacity of ROS may be altered by mitochondrial localization. Since ROS are molecules of short duration, the location of their production or signaling site can increase their efficiency.

Conventionally, complex I and complex III, including complex II, are considered the major contributors to ROS production [103]. However, the relative contribution of each site to the total production of $O_2^{\bullet-}$ and $H_2O_2$ varies from one organ to another and depends on respiration rate and redox state [19,44]. The different sites of ROS production have distinct signaling roles and presumably change under different physiological conditions [103]. It is therefore difficult to pinpoint the specific site of ROS production [104]. Up to eleven distinct mitochondrial sites of production of superoxide and/or hydrogen peroxide linked to substrate catabolism, electron transport and oxidative phosphorylation were recently identified in mammalian mitochondria [19,104]. Sites I [105] and III [44] are considered to generate predominantly or exclusively superoxide. Site II may generate both superoxide and hydrogen peroxide [103]. These sites may also act as sources of mitochondrial redox signal. $H_2O_2$ is the primary form of ROS utilized for intracellular signaling. Individual sites of ROS production are implicated in specific pathologies. Parkinson's disease and longevity are linked to superoxide production from the flavin- and ubiquinone (Q)-binding sites of respiratory complex I [106], ROS from the complex II flavin is linked to Huntington's disease and cancer [107,108] and ROS from complexes I, II and III and from mitochondrial glycerol phosphate dehydrogenase and matrix dehydrogenases are all invoked in ischemia/reperfusion injury [109,110].

Since most ATP is produced by mitochondria, impaired mitochondrial function is implicated in a variety of health chronic conditions and degenerative diseases [111], many of which can be attributed to excessive mitochondrial production of ROS. However, modest levels of ROS stimulate essential biological processes, such as proliferation, differentiation and immunity [112]. Furthermore, mitohormesis [113], a decrease in the net basal metabolism production of ROS, which increases resistance to oxidative stress [112], may be a way to improve mitochondrial function and resistance to chronic and degenerative diseases. Mitohormesis, a defense mechanism, can therefore promote health and increase longevity through the prevention or delay of diseases [113,114].

### 2.2.2. Peroxisomes

In peroxisomes, the respiratory pathway involves the transfer of electrons from various metabolites to oxygen, leading to the formation of $H_2O_2$ with the release of free energy in the form of heat [115]. It is not coupled to the production of ATP by oxidative phosphorylation [116]. Other free radicals produced in peroxisomes include $O_2^{\bullet-}$, $^{\bullet}OH$ and $NO^{\bullet}$. β-oxidation of fatty acids is the main metabolic process producing $H_2O_2$ in peroxisomes. However, different peroxisomal enzymes, such as acyl CoA oxidase, D-amino acid oxidase, L-a-hydroxy oxidase, urate oxidase, xanthine oxidase and D-aspartate oxidase, have been shown to produce different ROS [117]. Peroxisome and β-oxidation alterations are involved in many conditions and diseases, such as neurological disorders, and in the development of cancer [118].

### 2.2.3. Endoplasmic Reticulum

The electron transport chain of the endoplasmic reticulum is the second greatest source of ROS [94]. Catabolism of cell and foreign chemicals by cytochrome P450 includes redox steps and is responsible for the production of ROS in the endoplasmic reticulum. The enzymes of the endoplasmic reticulum that contribute to the formation of ROS include cytochrome P450, b5 enzymes and diamine oxidase [119]. Another important thiol oxidase, Erop1p, catalyzes the transfer of electrons from dithiols to molecular oxygen, resulting in the formation of $H_2O_2$ [120]. Other endogenous sources of ROS include the auto-oxidation of adrenalin, reduced riboflavin, inflammation, mental stress, over-exertion, infection, cancer, aging and ischemia [119].

### 2.2.4. Role of the Enzyme System

A variety of oxidative enzymes that occur in cells can produce free radicals. Those catalyzing ROS generation include nitric oxide synthases, NADPH oxidase, prostaglandin synthase, xanthine oxidase, lipoxygenases, ribonucleotide reductase, glucose oxidase, myeloperoxidase, cyclooxygenases and cytochrome P450 [28]. A certain quantity of ROS is produced by various oxidases. For example, xanthine oxidase and cytochrome P450 reductase mainly produce the superoxide anion radical, while oxidases of amino acids and glucose mainly generate hydrogen peroxide [121]. In particular, under normal physiologic conditions, xanthine oxidase acts as a dehydrogenase, removing hydrogen from xanthine or hypoxanthine and binding it to nicotinamide adenine dinucleotide (NAD), thus generating the NADH.

Lipoxygenase generates free radicals; it can convert PUFA to hydroperoxides once $Fe^{2+}$ has been oxidized to $Fe^{3+}$ [122]. The three major mammalian lipoxygenases are 5-, 12-, and 15-lipoxygenase; they can oxidize arachidonic acid, abundant in the central nervous system, to hydroperoxyeicosatetraenoic acid. In addition, 15-lipoxygenase has been identified in atherosclerotic lesions, suggesting that the enzyme may be involved in the formation of oxidized lipids in vivo [24].

High ROS levels are also generated by immune cells (lymphocytes, granulocytes and phagocytes) which defend the body against invading microorganisms [123]. Macrophages and neutrophils contain NADPH oxidase complex, which, when activated, generates superoxide radicals and hydrogen peroxide. The latter then interacts with intracellular chloride ions to produce hypochlorite, which destroys the pathogen. Patients with chronic granulomatous disease, in which ROS production is drastically reduced by NADPH oxidase complex, are highly sensitive to infections and usually die at an early age [124]. The main enzyme expressed by neutrophils is myeloperoxidase. With heme as a cofactor, it produces hypochlorous acid from hydrogen peroxide and chloride anion [125]. It also oxidizes tyrosine to the tyrosine radical. Hypochlorous acid and the tyrosine radical are both cytotoxic and are used by neutrophils to kill pathogenic organisms [126].

Cytochrome P450 molecules use $O_2$ in their biochemical reactions and generate small amounts of ROS. The amount of ROS produced varies depending on the compound degraded and the cytochrome P450 molecule involved. A molecule particularly active in the production of ROS is cytochrome P450 2E1 [127].

### 2.2.5. Role of Metals

The production of free radicals through reactions mediated by transition metals is well established [10,27]. Almost all transition metal ions have the ability to function in various oxidation states. In the active redox state, these ions may act as catalysts in the autoxidation of many biomolecules. In most cases, the oxidation of biomolecules is initiated by the hydroxyl radical (HO$^\bullet$) generated in Fenton and Fenton-like reactions between redox-active transition metal ions and hydrogen peroxide [128]. In biological systems, a two-step reaction may occur in the presence of metal ions, especially free iron, more important because of its abundance in biological material, or copper, leading to the production of hydroxyl radicals. Hydrogen peroxide can produce the hydroxyl radical by removing an electron from the participating metal ion [10]. In the second step, the superoxide radical is involved in regenerating the original metal ions, making them newly available for the reaction with hydrogen peroxide. The two chemical reactions support the role of metals such as iron and copper in creating oxidative stress and cell injury by ROS. Again, the redox state of the transition metal is more important for pro-oxidant activity than its concentration. The ferrous ion ($Fe^{2+}$) is a stronger pro-oxidant than the ferric ion ($Fe^{3+}$), which only shows pro-oxidant activity in the presence of a reducing agent, such as ascorbic acid [10]. The pro-oxidant activity of transition metals includes the decomposition of lipid hydroperoxides into free radicals capable of initiating or propagating lipid peroxidation [129]. The metals can decompose hydroperoxides to peroxyl and alkoxyl radicals and greatly accelerate lipid oxidation [66]. The ferric and ferrous ions can both be catalysts in the degradation of lipid

hydroperoxides to hydroperoxide-derived free radicals, but the catalytic activity of the ferrous ion is superior to that of the ferric ion. Moreover, the alkoxy radical is more reactive in the abstraction of a labile hydrogen atom than the peroxyl radical [27]. Because of the fundamental contribution of iron to the formation of hydroxyl radicals, any increase in cell concentration of free iron promotes the generation of ROS and oxidative stress [130].

### 2.3. Detection of ROS and RNS

The formation of ROS and RNS can be monitored by a variety of procedures, including fluorometric and spectrophotometric methods, chemiluminescence and electron paramagnetic resonance [44]. Many of these methods are based on the redox properties of specific ROS or RNS and are therefore subject to artifacts caused by species of similar reactivity or by reactive intermediates produced by the probe itself [131]. Electron paramagnetic resonance (EPR) spectroscopy has been studied to measure ROS, RNS and their secondary products [132]. The method is very suitable for the direct detection of free radicals at concentrations up to 1 $\mu$M. Due to its low sensitivity, EPR can measure ROS directly in vivo. It differs from other methods by virtue of its unique ability to detect free radicals with short and long half-lives, and it can provide information on oxygen/nitrogen radicals and related processes. Since NO is a free diatomic radical, it can be detected directly by EPR, even in tumors [133].

### 2.4. Molecular Targets of Free Radicals

The term reactive is not always appropriate for radical species; hydrogen peroxide ($H_2O_2$), nitric oxide ($NO^{\bullet}$) and superoxide ($O_2^{\bullet -}$) react promptly with some molecules, while the hydroxyl radical ($^{\bullet}OH$) reacts promptly with almost everything. Species such as peroxyl radicals ($RO_2^{\bullet}$), nitrate radicals ($NO_3^{\bullet}$), alkoxyl radicals ($RO^{\bullet}$), hypochlorous acid (HOCl), hypobromous acid (HOBr), carbonate ($CO_3^{\bullet -}$), carbon dioxide radicals ($CO_2^{\bullet -}$), nitrogen dioxide ($NO_2^{\bullet -}$), peroxynitrite ($ONOO-$), nitrogen dioxide ($NO_2^{\bullet}$) and ozone ($O_3$) have intermediate reactivities [10]. Free radicals have several types of reactions and targets. Oxidative processes are chemical/biochemical reactions involving the transfer of one or more electrons from an electron donor (reducing agent) to an electron acceptor (oxidizing agent). In biological systems, most molecules are non-radicals. Thus, when a free radical reacts with a non-radical, it creates a new radical and can trigger a chain reaction. When two free radicals meet, their unpaired electrons can unite to form a covalent bond. For example, $NO^{\bullet}$ and $O_2^{\bullet -}$ react rapidly to form $ONOO-$ [67], a non-radical product that rapidly protonates to peroxynitrous acid (ONOOH), a powerful oxidizing and nitrating agent that can directly damage proteins, lipids and DNA. Systems that produce $NO^{\bullet}$ and $O_2^{\bullet -}$ can therefore cause biological damage, linked to different human diseases [10,67]. Free radicals exist in low concentrations ($10-4$ to $10-9$ M), and their effects are observed locally where they are generated. Free radical reactions can chemically modify surrounding compounds, promoting the loss of physiological function in living organisms. Imbalance between the production of free radicals (ROS/RNS) and antioxidant defenses increases their concentrations, damaging important classes of biological molecules, such as nucleic acids, proteins, lipids and sugars (Figure 2), resulting in cell and tissue lesions with pathological implications [11,86].

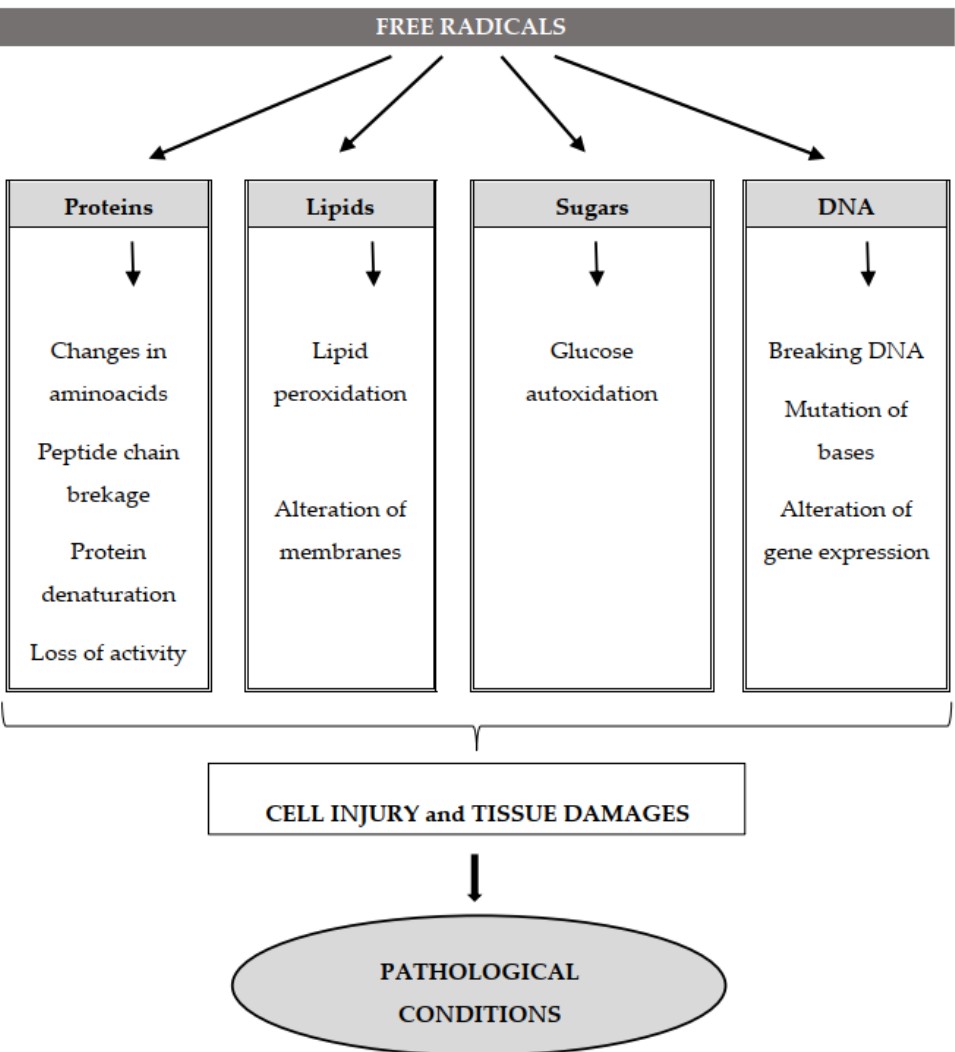

**Figure 2.** The molecular target of free radicals and oxidative damages. An excessive production of ROS and/or RNS can affect the main cell molecules and result in the oxidation of proteins, with alterations of amino acids, breakdown of the peptide chain, enzymatic inactivation, denaturation of proteins and loss of their activity; lipid peroxidation and alteration of membrane functions; autooxidation of glucose; fragmentation of DNA filament, base mutation and alteration of gene expression. All this leads to cell injuries, tissue damage and subsequent pathological disorders.

2.4.1. Deoxyribonucleic (DNA) and Ribonucleic (RNA) Acids

Both ROS and RNS can damage nucleic acids. The former damage DNA, inducing base modifications, deletions, strand breakage, chromosomal rearrangements, hyper- and hypo-methylation and gene expression modulation [31,134]. Mitochondrial DNA is more vulnerable to ROS attack than nuclear DNA because it is near where ROS are generated. A cell produces around 50 hydroxyl radicals per second; in a day, all the cells of a human would generate four million hydroxyl radicals, which can attack biomolecules [25]. The •OH radical reacts directly with all components of DNA, such as purine and pyrimidine bases and the deoxyribose sugar backbone [10,31], causing alterations, including breakages to single and double strands. It abstracts hydrogen atoms, modifying purine and giving rise to pyrimidine derivatives and DNA protein cross-linking [31]. It oxidizes guanosine and thymine in 8-hydroxyl-2-deoxyguanosine and thymine glycol, respectively, modifying DNA and triggering mutagenesis and carcinogenesis. In addition, 8-Hydroxy deoxyguanosine (8-OHdG) is considered a biomarker of DNA oxidative damage and has been used as a biomarker of oxidative stress [135]. It is also involved in mutagenesis, carcinogenesis and aging. Levels of 8- OHdG are higher in mitochondrial than nuclear DNA [10]. Reactive

nitrogen species, especially peroxynitrite, interact with guanine to produce nitrative and oxidative DNA lesions such as 8-nitroguanine and 8-oxodeoxyguanosine, respectively [136]. The resulting 8-nitroguanine is unstable and can be spontaneously removed, giving rise to an apurinic site; conversely, adenine can couple with 8-nitroguanine during DNA synthesis, resulting in G-T transversions. Therefore, 8-Nitroguanine is a mutagenic DNA lesion involved in carcinogenesis [135].

Reactive oxygen species can attack different RNAs produced in the body. RNA is more susceptible to oxidative damage than DNA, due to its single strand, lack of an active repair mechanism when oxidized and less protection by proteins. These cytoplasmic RNAs are located near mitochondria where ROS are abundantly produced [137]. Levels of 7,8-dihydro-8-oxo-guanosine (8-oxoG), the most studied product of RNA damage, are elevated in pathological conditions such as Alzheimer's disease [138], Parkinson's disease [139], atherosclerosis [140], hemochromatosis [141] and myopathies [142].

### 2.4.2. Lipid Oxidation

Lipid peroxidation is a physiological process that takes place in all aerobic cells. It is estimated to involve about 1% of all the oxygen taken up by cells, organs and bodies. The main targets of oxidation are polyunsaturated fatty acids (PUFAs) and phospholipids [143]. Free fatty acids have a carbonyl oxygen (C=O), deficient in electrons, and unsaturated fatty acids are deficient in electrons at their unsaturated carbon–carbon bonds (C=C); these regions of electron deficiency make fatty acids susceptible to attack by a variety of oxidizing agents [144]. Thus, lipid peroxidation takes place where there are high concentrations of polyunsaturated fatty acids. Triglycerides contain linear-chain fatty acids, mainly with 16- to 18-carbon atoms and minimal amounts of unsaturated fatty acids, while phospholipids in the tissue membranes contain up to 15 times the amount of unsaturated fatty acids (C18:4, C20:4, C20:5, C22:5 and C22:6) of triglycerides and are much more susceptible to oxidation due to the greater number of unsaturated carbon–carbon bonds (C=C) [145]. Polyunsaturated fatty acid residues in cell membranes are the main target of oxidation by free radicals.

A reactive radical (e.g., $^\bullet OH$ or $NO_2^\bullet$) may abstract a hydrogen atom from a C–H bond in a hydrocarbon side chain of a polyunsaturated fatty acid (PUFA) residue in a membrane. The resulting carbon-centered radicals ($C^\bullet$) react rapidly with $O_2$ to generate peroxyl radicals ($C–OO^\bullet$) that can attack side chains adjacent to PUFAs, triggering a chain reaction that forms lipid hydroperoxide. Hence, lipid peroxidation is initiated when any free radical attacks and abstracts hydrogen from a methylene group ($CH_2$) in a fatty acid (L:H), resulting in a carbon centered lipid radical ($L^\bullet$), which in turn reacts with molecular oxygen to form a lipid peroxyl radical ($LOO^\bullet$). The resulting lipid peroxide radical ($LOO^\bullet$) undergoes rearrangement through a cyclization reaction to form endoperoxides, which, in the end, form malondialdehyde (MDA) and 4-hydroxyl-nonenal (4-HNA) [12,142,146], recognized as markers of lipid oxidative decay. In particular, the cellular non-enzymic peroxidation of lipids, similar to all radical reactions, can be divided into three steps: initiation, propagation and termination. It proceeds as a chain reaction mediated by carbon- and oxygen-centered radicals [64]. The initiator of lipid peroxidation (usually an iron-oxygen complex) first produces a carbon-centered radical that reacts with oxygen. After the addition of oxygen, the lipid hydroperoxyl radical formed starts the propagation phase. The major mechanism of initiation is the abstraction of hydrogen by a lipid alkoxy radical, in turn produced by the breakdown of the O–O bond of a lipid hydroperoxide (LOOH). The players crucial for initiation are therefore a transition metal and a preexisting LOOH. In cells, this is achieved with a transition metal iron, which reacts by "Fenton chemistry" with hydrogen peroxide or superoxide produced endogenously to form oxygen-centered radicals [64]. In Fenton chemistry, ferrous iron disproportionates hydrogen peroxide, and the oxidant, ferric iron, generates an anion hydroxide and a highly reactive hydroxyl radical. If another equivalent of hydrogen peroxide is available, ferric iron can be reduced to its ferrous state and generate a peroxyl radical. Once the initial step is completed, lipid peroxidation

proceeds to the propagation step. Iron and copper can accelerate lipid peroxidation [10]. They convert $H_2O_2$ to $^{\bullet}OH$ by dividing the O–O bond (by Fenton reaction, in the case of $Fe^{2+}$). In a similar reaction, lipid hydroperoxides can be divided, giving radical alkoxyl ($LO^{\bullet}$) and more peroxyl ($LOO^{\bullet}$) to maintain the chain reaction. The free radical chain reaction propagates until two free radicals conjugate and terminate the chain. Enzymic lipid peroxidation by lipoxygenase, cyclooxygenase or cytochrome P450 occurs via regulated mechanisms to produce regio-, stereo- and enantio-specific products [147].

Cholesterol Oxidation

Cholesterol, (3β)-cholest-5-en-3-ol, is a lipid produced endogenously but can also be acquired from dietary sources. It plays important biological roles. The cholesterol content of membranes of eukaryotic cells has been estimated to range from 20% to 50% [148]. It is crucial for maintaining the stability, fluidity and permeability of cell membranes, but being a polycyclic monounsaturated alcohol, it is susceptible to oxidation, and its products are known as oxysterols. Oxysterols are 27-carbon derivatives of cholesterol created by enzymic or radical oxidation [149]. Several enzymes of the cytochrome P450 family oxidize cholesterol to specific hydroxyl cholesterols (OHCh), such as 7α- and 7β-hydroperoxycholesterol (7α- and 7β-OOHCh), 7α-OHCh, 7β-OHCh, 5α,6α-epoxycholesterol, 5b,6β-epoxycholesterol and 7-ketocholesterol (7-KCh) [150]. The oxysterols occur in vivo in different forms, namely esterified, sulfated, conjugated and free [149]. In very low concentrations, oxysterols mediate many physiological functions; they participate in the regulation of cholesterol metabolism, biosynthesis of bile acids and various signaling pathways [150,151]. Oxysterols can affect membrane fluidity, membrane permeability to cations, albumin and glucose and membrane-bound protein kinase c activity [151]. The accumulation of cholesterol oxidation products is mainly due to the overconsumption of cholesterol-rich foods, such as meat, eggs and dairy products [152], and has been associated with the onset and development of diseases, such as atherosclerosis, diabetes, Alzheimer's and Parkinson's disease [153], carcinogenesis and cancer progression [154].

Lipid Peroxides in Disease and Death

Lipid peroxidation is important because of its involvement in various pathological conditions. It affects cell membrane integrity (rupture of vacuolar or central lysosomal membranes), increases loss of membrane sealing against substances ($K^+$ and $Ca^{2+}$) that normally do not cross it except through specific channels and damages membrane protein inactivating receptors, enzymes and ion channels [155]. Lipid peroxidation involves the production of a variety of breakdown products, including alcohols, ketones, alkanes, aldehydes and ethers. About 32 aldehydes have been identified as products of lipid peroxidation, 4-hydroxynonenal (4-HNE) and malondialdehyde (MDA) being the best described and used as markers of lipid oxidation [156]. HNE is the major decomposition product of hydroperoxides of n-6 fatty acids. MDA and 4-HNE damage DNA, causing mutagenic lesions [143] and proteins. Both bind avidly to membrane proteins, inactivating enzymes and receptors. Changes in proteins from aldehyde products contribute to neurodegenerative diseases, the activation of kinases and the inhibition of nuclear transcription factors [157].

The toxicity of lipid peroxidation products in mammals generally involves hepatotoxicity, neurotoxicity and nephrotoxicity [64]; increases in products of lipid oxidation occur in liver diseases, diabetes, inflammation, atherosclerosis, cancer, apoplexy and aging. PUFAs and cholesterol molecules are a major part of the low-density lipoproteins (LDLs), and the oxidative modification of LDLs is reported to be involved in the development of atherosclerosis and cardiovascular disease. Lipid peroxides have long been considered important in the progression and regulation of inflammation [158]. Lipid peroxidation also has a role in neurodegenerative diseases [159], ischemia-reperfusion injury, diabetes [160] and mutagenic and carcinogenic activity [161].

Lipid peroxidation also plays a role in regulated cell death. The product 4-HNE has been shown to induce apoptosis in specific contexts [162], and lipid peroxidation is also the primary driver of ferroptosis, a type of regulated necrotic cell death [163]. However, it has been reported that many products of lipid peroxidation may have opposite effects, such as cytotoxic and cytoprotective, pro- and anti-inflammatory, pro and anti-atherogenic and pro and anti-apoptotic [164].

The assay of malondialdehyde (MDA) levels is one of the main methods of assessing the effects of ROS on biological systems. One rather non-specific method is based on the reaction of MDA with thiobarbituric acid (TBA) to form thiobarbituric acid reactive substances (TBARS), whereas an HPLC method is preferred if possible [165]. Lipid peroxides can be measured with different techniques, including the quite suitable iron/xylenol method [166]. Products such as MDA and 4-hydroxy-nonenal (4-HNE), which conjugate with proteins, can be measured by immune assay [167].

### 2.4.3. Protein Oxidation

Protein oxidation can be induced by radical species such as superoxide, hydroxyl, peroxyl, alkoxyl and hydroperoxyl or non-radical species such as hydrogen peroxide, ozone, hypochlorous acid, singlet oxygen and peroxynitrite anion [168]. Reactive oxygen species may attack proteins and produce carbonyls and other amino acid derivatives. The availability of oxygen, superoxide anion and its protonated form ($HO_2^-$) determines the process of protein oxidation. The induction of 3-chlorotyrosine from tyrosine by hypochlorous acid, the oxidization of histidine to 2-oxohistidine at the site of the metal binding of proteins, the oxidation of thiol groups and the generation of carbonyl derivatives of amino acids are some examples of protein modifications [57]. Reactive oxygen species cause oxidative damage to amino acid residues such as lysine, proline, threonine and arginine, producing carbonyl derivatives. The oxidation of different amino acids denatures proteins and causes loss of function, be it enzyme activity, receptor function or transport function [169]. Carbonyl groups on proteins were considered markers of oxidation proteins mediated by free radicals [170]. Specific markers of protein oxidation are O-tyrosine (a marker of the hydroxyl radical) and 3-nitrotyrosine (a marker of RNS). An increase in serum concentration of protein carbonyls is observed in various pathological conditions such as Parkinson's disease [171], muscle dystrophies [172], rheumatoid arthritis [173], progeria, atherosclerosis, Werner's syndrome [57] and aging [174]. Levine and Stadtman postulated an increase in protein carbonylation levels as a function of age [175]. A recent protein database lists over 180 proteins that are carbonylated during aging and in age-related disorders [176].

### 2.5. Free Radicals and Diseases

The implications of free radical processes concern the fields of biology and medicine. Free radicals accumulate due to an imbalance between antioxidants and oxidants that damage macromolecules, such as nucleic acids, proteins and lipids, causing abnormal gene expression, disturbance of receptor activity, cell proliferation, immune perturbation, mutagenesis, tissue damage and various disease conditions [87,177]. Many clinical disorders have been linked to free radicals (Figure 3), including diabetes mellitus [178], inflammatory diseases [179], neurodegenerative diseases (Alzheimer's [180], Parkinson's [181] and Huntington's disease [182]), lateral amyotrophic [183] and multiple sclerosis [184], cancer (colorectal [185], breast [186], prostate [187] and lung [188]), cardiovascular disease [189] (atherosclerosis [190] and hypertension [191]), cataract [192,193], rheumatoid arthritis [194], asthma [195] and aging [174].

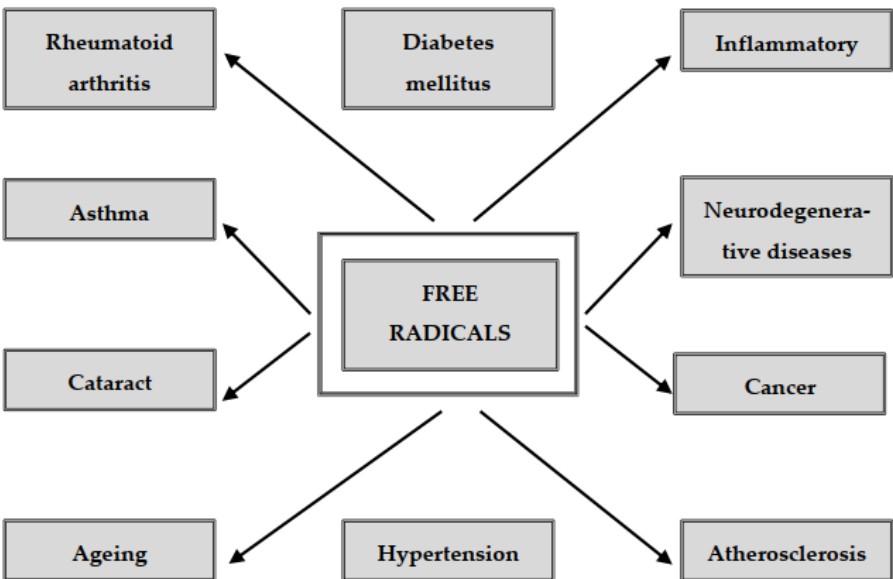

**Figure 3.** Free radicals and diseases. Under pathological conditions, an excess amount of free radical generation can cause protein oxidation, lipid peroxidation, DNA damage, changes in signal transduction and gene expression and several diseases.

### 3. Antioxidants

The deleterious effects of excess free radicals, or oxidative stress, have been reported to eventually lead to cell death. The overproduction of reactive oxygen and nitrogen species has been implicated in the development of various chronic and degenerative diseases [177]. The body has natural antioxidant defenses against free radicals. Antioxidants inhibit the oxidation process, even at relatively low concentrations, and can protect cells against free-radical damage by delaying or preventing the oxidation of proteins, carbohydrates, lipids and DNA [196]. Antioxidants have the ability to donate electrons. Antioxidants that break the chain reaction are strong electron donors and react with free radicals before major molecules are damaged. The antioxidants are therefore oxidized and must be regenerated or replaced. Antioxidant enzymes catalyze the degradation of species of free radicals, generally in the cell. Transition metal binding proteins prevent the interaction of transition metals, such as iron and copper, with hydrogen peroxide and superoxide and hence the production of highly reactive hydroxyl radicals.

Antioxidants can be endogenous or exogenous (Figure 4). The former may be enzymes, such as superoxide dismutase (SOD), catalase (CAT), glutathione peroxidase (GPx) and glutathione reductase (GRx), or non-enzymes such as metabolic antioxidants, such as lipoic acid, glutathione, L-arginine, uric acid, bilirubin and antioxidant nutrients [11]. Some exogenous antioxidant nutrients cannot be produced by the body and must be obtained from food; they include vitamin E, vitamin C, trace elements (Se, Cu, Zn, Mn) and phytochemicals such as isoflavones, polyphenols and flavonoids [197]. Endogenous and exogenous antioxidants are effective scavengers of free radicals; by giving electrons to ROS, they neutralize the negative effects of the latter, reducing oxidative stress and the oxidation of cell molecules [196]. The active interaction of endogenous and exogenous antioxidants has also been demonstrated; indeed, vitamin E administered orally to humans induces a significant increase in red blood cell glutathione levels. These findings may be explained by an increase in glutathione synthesis by stimulation of glutathione synthetase activity. An alternative possibility is the reduced utilization of glutathione for the detoxification of free radicals. These two mechanisms could be effective in counteracting glutathione feedback of its own synthesis [198].

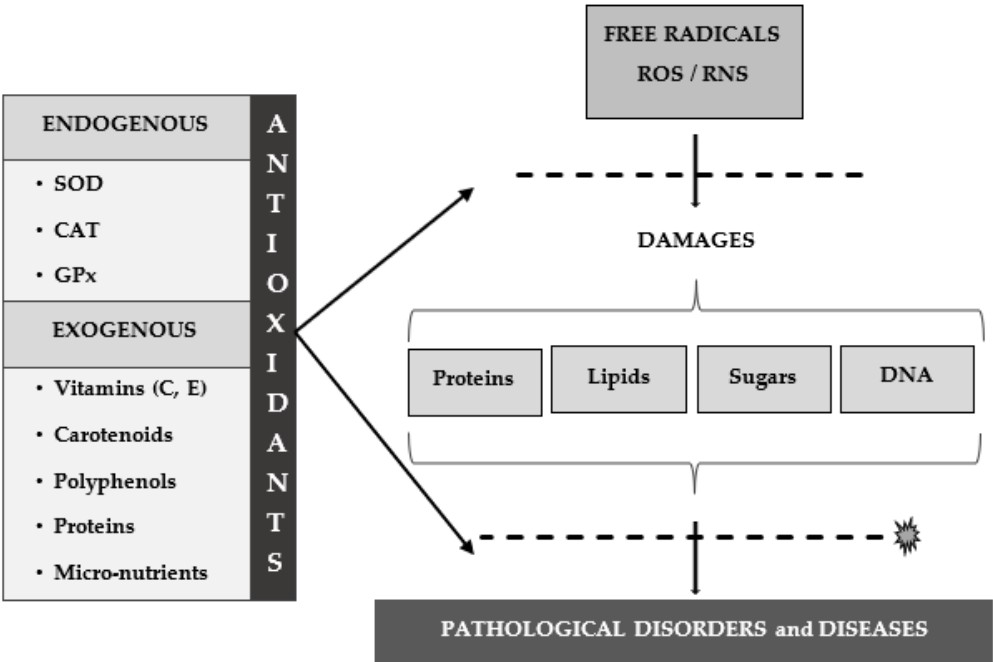

**Figure 4.** Antioxidant defense against free radicals. Antioxidants can be endogenous or exogenous. The antioxidants that break the chain are strong electron donors and react with free radicals before the major molecules are damaged. The first line of defense against free radicals includes the enzymes superoxide dismutase (SOD), catalase (CAT), glutathione peroxidase (GPx) and glutathione reductase (GRx). Exogenous antioxidants are compounds provided through foods or supplements, such as vitamin E, vitamin C, carotenoids, trace metals and polyphenols. These compounds represent the second line of defense against free radicals and consequent pathological disorders and diseases, acting with a scavenging effect, a stimulatory effect on endogenous antioxidant enzymes or both.

In recent years, the attention to diet as an essential source of exogenous antioxidants has increased. The demand for natural antioxidants has reduced the use of synthetic antioxidants due to their toxicity, carcinogenicity or hepatotoxicity in the human body [66]. The dietary antioxidants naturally present in food have aroused considerable interest because of their safety and potential nutritional and therapeutic effects on health [199]. Natural antioxidants can be found in all parts of plants such as fruit, vegetables, nuts, seeds, leaves, roots and bark [200], and people receive antioxidant supplements directly from fresh fruit and vegetables. It should be noted that, if antioxidant supplements are consumed in large doses, they can act as pro-oxidants [201]. Compared to supplements, the dietary intake of antioxidants from natural fruit and vegetables may be a safe way to avoid overdosing.

Foods rich in micronutrients such as $\alpha$-tocopherol (vitamin E) and minerals have been considered useful to alleviate ROS-related damage. For example, selenium and zinc interact with GPx and SOD, respectively, to combat oxidative stress. The combination of selenium and vitamin E has shown protective effects against oxidative damage in the colon of rats with ulcerative colitis [202]. Although dietary antioxidants are essential in supplying endogenous antioxidants for the neutralization of oxidative stress, inappropriate use may be detrimental for the physiological scavenging of ROS [203]. In fact, malnutrition and antioxidant deficiency have been related to diseases such as chronic obstructive pulmonary disease (COPD) and Crohn's disease [204,205]. The lack of antioxidants induced by malnutrition can increase the risk of disease and negative treatment outcomes [206]. Decreased intake or availability of dietary antioxidants such as vitamins C and E, carotenoids and polyphenols can reduce the efficiency of the antioxidant system and aggravate disease progression [204,207]. A diet rich in fruit, vegetables, fish and whole grains has been associ-

ated with better lung function and reduced risk of COPD [208]. Experimental evidence has suggested an association between nutrition and lens opacities.

A dietary deficiency of antioxidants and reactive oxygen scavengers may be involved in the pathogenesis of so-called "idiopathic" human senile cataract, as demonstrated in experimental cataracts. Indeed, the results of a pooled analysis showed a protective effect of antioxidants on the cataract, supporting the hypothesis of a nutritional deficiency in the human senile cataract, but not all studies reached statistical significance. Vitamin C, beta-carotene, lutein and zeaxanthin exert a protective effect, whereas the effect of vitamin E, vitamin A and alpha-carotene is not completely demonstrated [209].

The consumption of fruit and vegetables increases the blood concentration of antioxidants such as carotene and vitamin C and depresses the oxidation of cholesterol [210], and it has been inversely correlated with mortality due to CVD [211].

Epidemiological studies have detected a significantly reduced risk of CVD in relation to higher polyphenol intake [212]. Polyphenols are the most abundant antioxidants in the human diet (~1 g/d) and are common in fruit and vegetables, coffee, tea and cereals [213]. Their protective effects not only ensue from their antioxidant properties but also from their anti-inflammatory and vasodilatory effects on endothelial cells [214].

Clinical evidence indicates that deterioration in neurodegenerative diseases can be improved by the correct use of natural antioxidants [215]. It has been suggested that inadequate serum concentrations in vitamin D are highly associated with the loss of dopaminergic neurons in the brains of Parkinson's disease patients and with an increased risk of the disease [216]. Neurological impairment has also manifested in subjects with vitamin B deficiency. Multiple vitamin B deficiencies (B1, B3 and folate) are involved in the pathology of neurodegenerative diseases such as Parkinson's and Alzheimer's diseases [217]. In addition to vitamins, phytochemicals found in fruit and vegetables show high antioxidant capacity with potential neuroprotective effects against Parkinson's disease [218].

The aging population is at higher risk of malnutrition due to a general decline in body function, including reduced metabolic activity, digestive ability and absorption, as well as cognitive decline [219]. The elderly are therefore more likely to suffer from diseases associated with inadequate nutrition. Dietary antioxidants seem to play an anti-aging role due to their ability to suppress the generation of free radicals [220]. It has been found that women who consume more green leafy vegetables and crucifers have less cognitive decline [219]. A higher intake of green and yellow vegetables was also correlated with a slower rate of skin aging in Japanese women [221]. The supplementation of fruit and vegetable extracts significantly improves many parameters of immune function in elderly subjects [222].

The intestinal inflammation associated with oxidative stress plays an important role in various gastrointestinal diseases, such as inflammatory bowel disease [223,224]. Low levels of enzyme antioxidants and vitamins due to malnutrition were observed in subjects with inflammatory bowel syndrome [205]. Antioxidant use has been shown to restore redox balance, thereby mitigating intestinal damage and maintaining a healthy gastro-intestinal tract [223].

Fruit and vegetables, which are rich in antioxidants, exert a protective effect against several different types of cancer [225]. About 35% of cancers can be prevented by dietary modifications [226]. It has been demonstrated that plant foods containing polyphenols have anti-cancer activity that is effective against cancer of the lung, breast, tongue, stomach, larynx, colon and prostate [227,228]. Fruit with high phenolic content has stronger antioxidant properties that induce the replacement of the hydroxyl group in the aromatic rings of phenol compounds [229].

It should be emphasized that many dietary polyphenols are valid in cell experiments but are ineffective in animals and humans. For example, every year there are many reports of the anti-cancer activity of polyphenols on different tumor cell lines under controlled conditions [230]. However, few polyphenols are considered to be antitumor agents for

clinical use [231]. In the typical cytotoxicity or proliferation assay with dietary polyphenols and cancer cell lines, the single polyphenol or extract rich in polyphenols is only incubated with the cells for 24–72 h [232–234]. Again, quercetin, considered the most bioactive compound, has been claimed to inhibit cell growth and induce apoptosis, necrosis and autophagy in different cancer cell lines (A549 cells [235], HeLa cells [216], HT29 and HCT15 cells [236] and PATU-8988 cells [237]); however, quercetin is relatively insoluble in water and unstable in physiological systems, which leads to very low bioavailability [238,239]. There are no clinical studies demonstrating the efficacy of quercetin [240].

The lack of randomized trials on the relationship between diet and cancer is due to the difficulty of whole diet intervention for ethical reasons [241]. Thus, in order to reduce the risk of cancer, the general recommendations are a diet rich in fruit, vegetables and grains, a low intake of red meat and alcohol and a healthy lifestyle [241].

Thus, in regards to the use of antioxidants in the prevention and treatment of diseases, many problems still remain elusive. Some results indicate that antioxidants do not exert favorable effects on the control of the diseases. The precise roles of ROS in the pathogenesis of various diseases remain to be clarified. Most exogenously administered antioxidants are not selective or uniformly distributed in the various parts of cells or tissues [242,243]. The intake level (threshold level) of antioxidant nutrients needed for optimal nutrition is still unclear, and there is little information on antioxidant bioavailability in vivo in humans [244,245]. The lack of specificity of antioxidants and their possible interactions [245,246] may explain their ineffectiveness in the treatment of diseases related to oxidative stress. It is therefore suggested to concentrate on the development of innovative targeted antioxidants to focus their precise therapeutic effects [243,247].

### 4. Conclusions

Free radicals are produced by various normal endogenous and exogenous cell processes. Mitochondria are the main source of endogenous cellular ROS, which, in high concentrations, may damage macromolecules such as DNA, proteins and lipids. Such damage has been implicated in the development of various pathological disorders, including cancer and inflammatory, respiratory, cardiovascular, neurodegenerative and digestive-tract diseases. The body has innate antioxidant defenses against free radicals. Recent nutritional recommendations include a diet rich in fruit, vegetables and grains. Dietary antioxidants are considered useful in the prevention and treatment of disease, although some studies indicate that they are not particularly effective in the control of disease. Many questions regarding antioxidant supplementation and disease prevention remain unanswered. Further research is needed before antioxidant supplements can be recommended as adjuvant therapy.

**Author Contributions:** Conceptualization, writing and review of draft, G.M.; critical revision of manuscript, C.C.; literature review, writing of some paragraphs and editing, M.M., L.D. and P.N.; conceptualization, supervision, review and funding acquisition, A.G.D. All authors have read and agreed to the published version of the manuscript.

**Funding:** This study was funded by the Innovation and Competitiveness Cooperation Programme Interreg V/A Greece-Italy (EL-IT) 2014–2020 co-financed by the European Regional Development Fund (ERDF)—INNO.TRITION (Mis. Code: 5003778).

**Conflicts of Interest:** The authors declare no conflict of interest.

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
