# Peer review of "Free Radical Properties, Source and Targets, Antioxidant Consumption and Health"

_oxygen, doi:10.3390/oxygen2020006_

Round 1

Reviewer 1 Report

The manuscript entitled  “Free radical properties, source and targets, antioxidant consumption and health” by Martemucci et al, reviews the role of free radicals in the fields of biology and medicine, the consequences of their overproduction in biological materials and discusses the potential role of antioxidants in the prevention or therapy of diseases.

Minor comments

  1. Line 64. free radicals are species capable of independent existence that contains unpaired electrons, however they are not necessarily reactive.

  1. Formulas for radicals should follow IUPAC recommendations (a superscripted dot that precedes a charge, if any applies). Additionally, radicals as well as other species should be written in the same manner in the text.

I noticed  for example that superoxide anion is written as O2•−(which is correct) but also as O2−•, O2, O•-. Hydroxyl radical is writer as HO, OH and OH(the last one is not correct) etc.  

  1. Lane 108. The term ROS is referred to oxygen derived radicals and non-radical species, thus in my opinion the word “often” is not necessary.

  1. Lane 144. Superoxide is considered as a moderately reactive free radical in biological systems.

  1. Lane 151 in vitro should be written as in vitro.

  1. Lane 216

Oxygen accumulated in Earth’s atmosphere sometime between 2.5 and 2.3 billion years ago. (What we call the Great Oxidation Event)

  1. Lane 414 correct “hidrogen” to “hydrogen”

  1. Lane 418, 420. Write etc. instead of …

  1. Lane 429. A small proportion is converter to ROS (not only free radicals)

  1. Lane 569 (Role of metals)

Transition metals catalyze the formation of hydroxyl radical (Fenton reaction), however iron is more important, most likely because of its abundance in biological material.

  1. Correct in the text in figure 4: “enzyme” to “enzyme” , “glutathione” to “glutathione”

  1. Use “aging” or “ageing” (the one word or the other).

Author Response

Thanks the reviewer for the suggestions and corrections. The manuscript was modified accordingly.

Reviewer 2 Report

Title: Free radical properties, source and targets, antioxidant consumption and health

Review Feedback

Authors: Giovanni Martemucci et al

The authors are reporting, in this review, the main properties of free radicals, their sources and deleterious effects. They also focused on the potential role of dietary supplementation of  antioxidants and discussed unsolved problems regarding antioxidant supplements in the prevention and therapy of diseases.

  1. Introduction: the introduction does provide the reader the recent free radical history while also properly citing references.

  1. Free radicals: The first paragraph introducing oxygen as an important biological molecule does not seem to cited properly. There is only one in-text citation at the end of the paragraph after they make many statements about what makes oxygen so important in these systems. They cite a paper on free radicals in medicine, but this only seems to be in reference to lines 62-63, not the whole paper. After this, while describing the various reactivities of different common radical species, it would have been beneficial to include a figure directly ranking reactivity of them in the biological systems.

The authors included half lives in seconds but don’t directly say that this indicates their reactivity. The authors say that “despite its super name [superoxide]’s reactivity with biomolecules is low” (123-124) but then end this section of the text by saying that it is “highly reactive” (144). Perhaps this is a misunderstanding on my part but it seems like there wording could be phrased better to avoid this confusion.

The end of section 2.1.2 does a good job at citing sources of different damage that the hydroxyl radical can do. Later on, they reference the half-lives of the radicals and relate this to harm in that they can travel farther (281). This information would have been useful to highlight earlier in the paper so that the reader can go into the past 8 subsections with this understanding. This section also had some typos and issue with graphics: In line 185 “oarts” instead of “parts”, in line 212 “H2O2” instead of “H2O2”. Section 2.1.13 starts off awkwardly, with “The sulfur is…” (335), instead, this should just say “Sulfur is …”. Additionally, Figure 1’s spacing is not well managed and it results in an awkward looking figure.

Section 2.2.1  review the currently known information about ROS from mitochondria with abundant sources every other sentence. Section 2 overall was a great addition to the paper. They give in depth mechanistic insight into how ROS and RNS interfere with biological systems and what certain aspects of them need to be quenched to reduce pathology.

  1. Antioxidants: I appreciate that they explain the different types of antioxidants currently under consideration and that are already present in the cell.

Since many in vitro cell studies do not transfer to in vivo studies. They recommend the current status quo within the field of antioxidants to consume a high amount of dietary antioxidants, specifically referencing polyphenols.

Figure 4 in this section also has a mistake in its text, saying “patological” instead of “pathological” and they continue to use this misspelling in line 936.

Author Response

(The authors gave the same response as above.)

Reviewer 3 Report

This is a very interesting paper, well organized and well written.

However, some paragraphs need to be strongly improved and major revisions are required.

2.1.7 Ozone

the term 'ozonolysis' must be introduced as well as the impact of O3 on cholesterol which favors the generation of triol which is a strongly cytotoxic cholesterol oxide derivative. The consequence of O3 on lipid peroxidation and susbsequently on human health (civilization diseases) must be discussed. Additionnal references are required such as:

  • Iuliano L. Pathways of cholesterol oxidation via non-enzymatic mechanisms. Chem Phys Lipids. 2011 Sep;164(6):457-68. doi: 10.1016/j.chemphyslip.2011.06.006. Epub 2011 Jun 15. PMID: 21703250.

2.2.2 Peroxisomes

The role of peroxisome in the control of the RedOx status is very important in neurodegeneration. The first enzyme of peroxisomal beta-oxidation (ACOX1 is coupled with catalase). This paragraph is very short and must be strongly improved. Lot of refs must be added such as:

  • Singh I, Pujol A. Pathomechanisms underlying X-adrenoleukodystrophy: a three-hit hypothesis. Brain Pathol. 2010 Jul;20(4):838-44. doi: 10.1111/j.1750-3639.2010.00392.x. PMID: 20626745; PMCID: PMC3021280.
  • Trompier D, Vejux A, Zarrouk A, Gondcaille C, Geillon F, Nury T, Savary S, Lizard G. Brain peroxisomes. Biochimie. 2014 Mar;98:102-10. doi: 10.1016/j.biochi.2013.09.009. Epub 2013 Sep 21. PMID: 24060512.
  • Fransen M, Revenco I, Li H, Costa CF, Lismont C, Van Veldhoven PP. Peroxisomal Dysfunction and Oxidative Stress in Neurodegenerative Disease: A Bidirectional Crosstalk. Adv Exp Med Biol. 2020;1299:19-30. doi: 10.1007/978-3-030-60204-8_2. PMID: 33417204.
  • Ferdinandusse S, Finckh B, de Hingh YC, Stroomer LE, Denis S, Kohlschütter A, Wanders RJ. Evidence for increased oxidative stress in peroxisomal D-bifunctional protein deficiency. Mol Genet Metab. 2003 Aug;79(4):281-7. doi: 10.1016/s1096-7192(03)00108-2. PMID: 12948743.

2.4.3. Cholesterol oxidation

Triol is missing among the oxysterol cited whereas it is also formed by auto-oxidation. Please cite the following ref: Vejux A, Samadi M, Lizard G. Contribution of cholesterol and oxysterols in the physiopathology of cataract: implication for the development of pharmacological treatments. J Ophthalmol. 2011;2011:471947. doi: 10.1155/2011/471947. Epub 2011 Apr 4. PMID: 21577274; PMCID: PMC3090752.

The last sentence of the paragraph is not appropriated and must be rewritten. Indeed, cholesterol oxidation products are not mainly brought by food. They are aso the consequence of abnormal cholesterol metabolism and concerning the oxysterols formed by auto-oxydation from a ruptyre of RedOx status. The term oxysterome must be introduced. These remarks must be taken in consideration and this paragraph must be improved. Among the references which must be added, I indicate:

  • Mutemberezi V, Guillemot-Legris O, Muccioli GG. Oxysterols: From cholesterol metabolites to key mediators. Prog Lipid Res. 2016 Oct;64:152-169. doi: 10.1016/j.plipres.2016.09.002. Epub 2016 Sep 26. PMID: 27687912.
  • Guillemot‐Legris, O., Muccioli, G.G., 2021. The oxysterome and its receptors as pharmacological targets in inflammatory diseases. Br J Pharmacol bph.15479.https://doi.org/10.1111/bph.15479
  • Zarrouk A, Vejux A, Mackrill J, O'Callaghan Y, Hammami M, O'Brien N, Lizard G. Involvement of oxysterols in age-related diseases and ageing processes. Ageing Res Rev. 2014 Nov;18:148-62. doi: 10.1016/j.arr.2014.09.006. Epub 2014 Oct 14. PMID: 25305550.
  • Anderson A, Campo A, Fulton E, Corwin A, Jerome WG 3rd, O'Connor MS. 7-Ketocholesterol in disease and aging. Redox Biol. 2020 Jan;29:101380. doi: 10.1016/j.redox.2019.101380. Epub 2019 Nov 14. PMID: 31926618; PMCID: PMC6926354.
  • Zerbinati C, Iuliano L. Cholesterol and related sterols autoxidation. Free Radic Biol Med. 2017 Oct;111:151-155. doi: 10.1016/j.freeradbiomed.2017.04.013. Epub 2017 Apr 18. PMID: 28428001.

Figure3 must be modified. Aging is not a disease.The legend must be modified and the figure must be improved.

In addition, aging can be favored by civilization diseases and aging can favor age related diseases. In this context, the RedOx status plays an important role. Consequently, the paragraph 2.5 Free radicals and diseases must be improved.

Author Response

(The authors gave the same response as above.)

Round 2

Reviewer 3 Report

The review is know very clear and contains lot of information which will be usefull for lot of researchers.